# Improvement of Tiny Object Segmentation Accuracy in Aerial Images for Asphalt Pavement Pothole Detection

**DOI:** 10.3390/s23135851

**Published:** 2023-06-24

**Authors:** Sujong Kim, Dongmahn Seo, Soobin Jeon

**Affiliations:** 1Department of Computer Software, Catholic University of Daegu, Gyeongsan-si 712010, Republic of Korea; vhrxksro@cu.ac.kr; 2School of Computer Software, Catholic University of Daegu, Gyeongsan-si 712010, Republic of Korea; sarum@cu.ac.kr

**Keywords:** pothole, road maintenance, aerial image, UAV, instance segmentation

## Abstract

In this study, we propose an algorithm to improve the accuracy of tiny object segmentation for precise pothole detection on asphalt pavements. The approach comprises a three-step process: MOED, VAPOR, and Exception Processing, designed to extract pothole edges, validate the results, and manage detected abnormalities. The proposed algorithm addresses the limitations of previous methods and offers several advantages, including wider coverage. We experimentally evaluated the performance of the proposed algorithm by filming roads in various regions of South Korea using a UAV at high altitudes of 30–70 m. The results show that our algorithm outperforms previous methods in terms of instance segmentation performance for small objects such as potholes. Our study offers a practical and efficient solution for pothole detection and contributes to road safety maintenance and monitoring.

## 1. Introduction

Most countries have regulations and laws for the maintenance, safety inspection, and repair of roads built in accordance with their road laws. Suitable planning for maintenance and repair is crucial, as road safety is directly related to public safety. Road maintenance, safety inspections, and repair procedures can vary between countries. Road patrol and inspection are particularly important for ensuring the safety of vehicles, as the road surface condition can significantly impact vehicle safety.

Potholes are a major cause of road accidents, and many countries invest substantial efforts into repairing and preventing them [1]. Since maintaining road safety requires early detection and repair of potholes, pavement management systems are widely used [2]. Representative categories of methods for detecting potholes include direct inspection by a person and 3D laser scanning-based and image-based methods. The method of direct human inspection poses limitations in that it can only inspect up to 10 km per day [3].

The 3D laser scanning-based method is a technique used to detect potholes and other road surface damage. This method involves attaching a laser sensor to a vehicle and projecting a laser beam onto the road surface to create a 3D map. By analyzing the data from the 3D map, potholes and other types of damage can be detected in real-time [4]. 

The image-based method locates potholes by attaching a camera to a vehicle and analyzing the captured images of the road surface. However, using this method with a car has several limitations. First, the camera has a fixed field of view (FOV) at the front of the vehicle, which may not capture the opposite lane. Therefore, the vehicle is driven on the same road section in both directions to capture all potholes. Second, widening the angle of view is possible by attaching multiple cameras to the car. However, if there is a median on the road, it becomes important to drive on the same road section in both directions, similar to when only one camera is attached to the vehicle. As the image-based method using a car poses limitations, unmanned aerial vehicles (UAVs) may offer an alternative solution for detecting potholes on the road surface.

Using a UAV to photograph potholes on the road surface offers several advantages over using a car. First, UAVs can access relatively difficult locations compared to cars and can cover a wide area, eliminating the need to drive in both directions to capture the road surface. Second, by operating the UAV remotely, the operator’s safety can be ensured even if there are potholes during the road surface inspection. Third, although the cost of purchasing UAVs and camera equipment may be higher than attaching cameras to vehicles, the operating and maintenance costs can be considerably lower. Fourth, UAVs are cost-effective and require minimal human resources, making them practical and efficient for road maintenance and monitoring.

UAVs can access relatively inaccessible locations and a wide range of target areas. Because UAVs capture a wide range of objects, various objects exist in the captured images. To use an image captured by a UAV, identification of each object in the image is necessary. Object identification includes methods in which a person directly identifies the object and a method based on artificial intelligence (AI). Because humans recognize objects with the naked eye, their ability to distinguish objects differs according to the person’s experience. The different abilities to distinguish objects imply that people have many subjective opinions when deciding on object categories. Therefore, even the same object may be classified differently depending on the person, which is disadvantageous.

The performance of instance segmentation in a UAV environment is inferior for the following reasons. First, the result of synthesizing the performance evaluation results of instance segmentation models using the Common Objects in the COntext dataset (COCO) [5] shows that the AP (averaged over IoU thresholds) is overall 40% [6,7]. In addition, the evaluation results show that the ${AP}_{s}$ is the lowest among ${AP}_{s}$, ${AP}_{M}$, and ${AP}_{L}$, indicating that it is difficult for the instance segmentation model to infer the areas of small objects. Therefore, the low ${AP}_{s}$ of the instance segmentation model indicates that it is difficult to segment an object accurately in a small UAV environment. Second, the instantaneous field of view (IFOV) of a UAV is the angle that the UAV can capture in a moment. The altitudes of the IFOVs and UAVs determine the sizes of the resolution cells. The resolution cell is one pixel in the image. When photographing the ground using a UAV, the size of the object on the ground must be larger than the size of one pixel to distinguish the object. In addition, in a UAV environment, the number of pixels occupied by each object decreases as the altitude increases. Therefore, it is challenging to accurately infer small objects.

To solve the problem of low instance-segmentation performance in a UAV environment, Han applied an image-processing method to the object detection results of an AI model [8]. By utilizing the image-processing-based problem-solving method, Han attempted to solve the problem by applying an edge filter to the object detection result of the Mask R-CNN [9] and connecting the detected pixels using a neighborhood algorithm. 

In this study, we propose an image-processing-based algorithm to improve the instance segmentation performance of small objects, specifically potholes, by addressing the limitations of the algorithm proposed by Han. In addition, we experimentally evaluated the performance of the proposed algorithm. For the experiment, we filmed roads in various regions of South Korea using a UAV at high altitudes of 30–70 m. The proposed algorithm uses Mask R-CNN to set the initial range, detects the contour using various edge detection algorithms, and performs denoising. The VAPOR process decides whether to apply it and derives the result by connecting the detected pixels. In summary, the main contributions of the paper are twofold. First, Mask R-CNN generates the initial contour, and the proposed algorithm is not needed to set the initial contours manually. Second, Experiments on high-altitude captured images show the robustness of the proposed method compared with tested Mask R-CNN, ACMs.

The remainder of this paper is organized as follows. Section 2 describes a review of Active Contour Models (ACMs) and Han’s image-processing-based algorithm while also discussing the relevant problems. In Section 3, an algorithm to overcome the limitations of Han’s algorithm is discussed. Section 4 presents the measurement and evaluation of the performance of the proposed algorithm. Finally, Section 5 presents the conclusions of this study.

## 2. Related Work

### 2.1. Active Contour Model

Image segmentation plays a vital part in computer vision, and in a system using this technique, image segmentation results directly affect the performance of the whole system [10]. In recent years, the active contour model (ACM) based on partial differential equation (PDE) has achieved significant success among the numerous image segmentation algorithms proposed [11].

Existing ACMs can be classified into two classes: edge-based models and region-based models [12]. The edge-based model performs well on images with clear boundaries. However, unsatisfactory with blurred boundaries and can become stuck in local minima due to noise and texture. The region-based model is robust to noise and initial contour position and effectively handles images with weak edges, allowing for accurate and reliable segmentation results. However, it faces challenges with images containing non-uniform gray levels in the target or background areas [13].

Chen [14] contrasted the performance of various ACMs with deep learning-based algorithms. The results indicated that some of the deep learning-based algorithms outperformed the ACMs. This superiority may stem from the inherent ability of deep learning algorithms to discern complex patterns and relationships, handle diverse types of noise, and maintain resilience in the face of varying image quality. Deep learning algorithms may provide a more robust and versatile solution for future image segmentation tasks.

Figure 1 shows the results of medical contours on images using ACM. Bias correction (BC) [15], Local image fitting (LIF) [16], Local pre-fitting (LPF) [17], and Region scalable fitting and optimized Laplacian of Gaussian (RSF&LOG) [18] show that each of the models has relatively accurate contours in medical images.

### 2.2. Existing Algorithm and Challenge

A pothole is a defect or damage that occurs on a road surface that poses a threat to drivers driving on the road and can cause damage to tires. Since it is important to detect and maintain these potholes early, various methods have been proposed for their detection. One method for detecting potholes is to identify potholes in aerial images captured by UAVs using AI. Han studied a system that detects potholes in aerial images using an instance segmentation model.

Figure 2 shows the architecture of Han’s pothole detection system. In addition to the pothole detection function, Han’s system exhibits an area measurement function to measure the pothole risk, a coordinate and north direction measurement function to determine the pothole location, and a function to visualize and output the detection result. We focused on the pothole-detection function in Han’s proposed system. Han et al. used a Mask R-CNN to detect potholes. For mask R-CNN learning, we directly photographed road potholes using a drone. Han built a dataset by labeling, as shown in Figure 3, using the CVAT [19] labeling tool. The only object that Han annotated and labeled was a pothole. Objects other than potholes, teas, and trees were not labeled.

The pothole dataset is composed of the following: 448 training, 138 validation, and 100 test datasets. In addition, using image processing-based image augmentation, such as affine transformation, image inversion, and Gaussian blur, the number of training and validation datasets was increased to 2000 and 500, respectively. In addition, Mask R-CNN training was performed by setting the epoch, that is, the number of learning iterations, to 160, and the steps per epoch, that is, the number of weight updates, to 1000.

As the size of the pothole increases, the driver’s risk increases because of factors such as tire tear [20,21]. Therefore, Han et al. proposed an image-processing-based algorithm to estimate pothole areas more accurately. Han’s proposed algorithm proceeds as follows: (1) Instance segmentation and bounding box extraction in aerial images using a Mask R-CNN, (2) Extraction of the outline of the extracted bounding box using canny edges, (3) Bitwise and operational progress of the extracted edge and instance segmentation, (4) Outline connection detected using the *K*-D tree [22], and (5) The inside of the outline is filled using a Flood Fill [23].

However, Han’s algorithm exhibits two drawbacks. First, Han’s algorithm performs a bitwise AND operation in (3). Due to the fact that the bitwise AND function causes the edges detected outside the instance segmentation to disappear, the object’s edge extraction is limited to the inside of the mask. Second, Han used two thresholds to extract pothole edges. Han’s algorithm does not detect an object’s edge outside the two thresholds set. To solve the above problem, an additional algorithm was designed to improve instance segmentation; however, the results obtained were inferior.

Figure 4 shows the results of pothole inference from aerial images using Han’s pothole detection model. The pothole instance segmentation result using the Mask R-CNN was relatively inaccurate compared with the actual pothole area. The segmentation was considerably larger than the actual pothole area and did not entirely cover the actual pothole area.

## 3. Improved Approach for Ensuring Object Segmentation Accuracy

Figure 5 shows a flow diagram of the proposed algorithm. The flow diagram of the proposed algorithm consists of three steps: Multiple Orientation Edge Detector (MOED), VAlidation Phase Of Results (VAPOR), and Exception Processing. First, MOED extracts only the edge of the actual pothole in the image. Second, VAPOR determines whether the MOED result is normal or abnormal based on the two defined threshold values. If the result determined through the VAPOR process is normal, we use the *K*-D tree to connect all points of the detected edge according to the set criteria. If the MOED result is abnormal, we proceed with Exception Processing.

Third, when proceeding with exception processing, the result of our proposed algorithm is significantly larger or smaller than the ground truth (GT). At this time, an image-processing-based algorithm for accurately estimating the pothole area proposed by Han was applied. Han’s algorithm exhibits two limitations (Section 2.2). However, when comparing the mask R-CNN instance segmentation and Han’s algorithm results with the GT, Han’s algorithm exhibits higher performance than instance segmentation. Therefore, we use the algorithm proposed by Han for exception processing.

### 3.1. Multiple Orientation Edge Detector (MOED)

The proposed algorithm conducts a test to target potholes in images acquired by UAVs at high altitudes. The pothole exhibits a significant shape change depending on the damage. Therefore, estimating the general morphology of damaged potholes is difficult. In general, the shape estimation of an object in a two-dimensional image detects edge candidates through differentiation. An edge is a point that shows a clear difference from the background among the detected candidate groups. Therefore, an edge indicates a pixel whose brightness value in an image changes rapidly. In the edge candidate group of objects detected through differentiation, a significant amount of noise exists along the edges of the objects. Distinguishing between edges and noise is difficult. Various methods are available for removing noise from these images. However, when applying the noise removal method to an image, there must be a clear difference in the value between the edge and noise. Otherwise, there is a risk that noise removal may remove the edge along with the noise. Therefore, before applying a noise removal method, the edge of the object in the image must be rendered more distinct to distinguish it from noise.

We provide Figure 6 for an overview. MOED is an algorithm for extracting only the edges of potholes from an image. MOED consists of two steps: edge detection and noise filtering. First, in the edge detection step, pothole edges are extracted using a Steerable Filter (SF) [24] and a Laplacian filter (LF) [25] and subsequently merged, as shown in (Figure 6①). Second, the noise filtering step removes noise within the image and clusters smaller than a specific size through DBSCAN [26] and minimum cluster size (MCS) filtering, as shown in Figure 6②. The performance of each algorithm in the MOED changes according to the parameter value. Therefore, we conducted an experimental evaluation to obtain apprOPRiate parameter values.

#### 3.1.1. Edge Detection

We applied SF to distinguish the edges of the object. SF is a convolutional kernel image-processing algorithm with a selectable orientation. The SF parameters are theta, which indicates the direction of the filter, and sigma, which is the size of the kernel. Sigma indicates standard deviation in the Gaussian filter. The standard deviation is the size of the Gaussian filter; the larger the standard deviation, the greater the degree of blurring.

Figure 7 shows the results according to the SF sigma and theta changes. The SF parameter theta value is applied in intervals of 15° from 0° to 360° to clarify the object’s edge in all directions. Because the sigma value of the SF determines the degree of blurring, experimentally obtaining a sigma value suitable for pothole detection is necessary. The edge of the object sharpened through SF was detected using an edge-detection algorithm. Edge detection algorithms include the Laplacian, Canny [27], Prewitt [28], Robert [29], Scharr [30], and Sobel [31]. Therefore, determining the most suitable algorithm for pothole edge detection is necessary.

Figure 8 shows the results of applying the LF to the SF results. The value of each pixel in the resulting image ranges from 0 to 255. Identifying only the edges in this relatively wide range of values is difficult. Therefore, we apply Otsu’s [32] binarization technique to distinguish the edges from noise. Because of the binarization, the images exhibit distinct edges in a specific direction. Therefore, merging the resulting images into one object is necessary. Figure 9 shows the merging results after applying the edge-detection filter to the SF result. Otsu binarization was performed only on the LF merge results. We compared the effects of each edge-detection filter and found that the LF exhibited the best results.

#### 3.1.2. Noise Filtering

Noise can occur regardless of the sigma value of SF and the type of edge-detection filter. When connecting pixels with a value of 255 in the image to obtain the area of the pothole, if noise other than that at the pixel edge of the pothole exists, the area of the pothole may increase rapidly. Therefore, the removal of noisy data is essential. Many algorithms for noise detection exist; however, we used DBSCAN, a data-density-based clustering algorithm.

Figure 10 shows the results of applying DBSCAN and MCS filtering to the edge image. The DBSCAN parameters used were an eps value of two and a minPts value of 20. DBSCAN was utilized to eliminate noise in the images and was found to be an effective method for noise removal.

DBSCAN uses the eps and minPts parameters to filter noise. The algorithms use these parameter values to determine clusters by checking the proximity of the data points. Each point checks the number of points within the eps radius and forms a cluster if it satisfies the minPts condition. An extremely large eps may form a single cluster containing all points, and extremely large minPts may not form a cluster, even if the value of eps is appropriate. In other words, the determination of the appropriate eps and minPts is necessary.

The resolution of the pothole image ranged from a minimum of 40 × 39 pixels to a maximum of 214 × 304 pixels. DBSCAN alone is not sufficient for removing noise from images of various resolutions. Therefore, we analyzed the clusters formed using DBSCAN. Our analysis was significant only when the cluster formed using DBSCAN exhibited more than a specific ratio of the total number of pixels in the image. We define the MCS equation for judging cluster significance as follows:(1)$$MCS=\frac{1}{{\mathrm{coeff}}^{2}}hw$$
where w (width) and h (height) are the pixel sizes of the image and are variables that change according to the image resolution. *Coeff* is a *coeff*icient used for the MCS calculation. *Coeff*, a *coeff*icient used in the MCS calculations, also requires an appropriate value for pothole detection through experiments.

### 3.2. Validation Phase of Results (VAPOR)

Figure 11 for VAPOR provides an overview of VAPOR. VAPOR verifies the result of connecting the pothole-edge pixels obtained using the MOED. Our algorithm assumes that the mask inferred by Mask R-CNN contains most of the edges of the pothole. As shown in Figure 11a, in case of a considerably large number of pixels outside the mask than inside, obtaining the desired result is difficult, even if the area is obtained by connecting all the detected points. Therefore, exception processing is performed in this case. As shown in Figure 11c, in case of a considerably large number of pixels inside the mask than outside, the probability of obtaining the desired normal result is high. The ratio between the pixels inside and outside the mask was calculated to distinguish between the normal/abnormal regions of the resulting image. The Outer Pixels Ratio (OPR), which is the ratio of the pixels outside the mask in the resulting image, is defined as follows:(2)$$OPR=\left\lfloor 100\frac{\left|\left(Mask\cup Img\right)-Mask\right|}{\left|img\right|}\right\rfloor$$

OPR represents the proportion of pixels outside the Mask in the resulting image. Mask and img are the results of processing through the mask and the MOED inferred by the Mask R-CNN, respectively. The result of an OPR operation is always between 0 and 100. Specifying the 101 values best for a pothole image is challenging. Therefore, we experimentally determined the most suitable OPR for the potholes. We calculated the OPR of the edge image obtained using the MOED. When the calculated OPR exhibits a value larger than the most appropriate OPR found experimentally, we judge it as a failure case and perform exception processing. If the OPR of the edge image acquired using MOED was less than the set OPR, we connected all detected pixels to obtain the area.

A route connecting each detected pixel is required to calculate the area of an object in the edge image. The area of an object may vary depending on the route connecting the pixels. In addition, according to MOED, there are as many routes as the number of pixels detected. Route creation begins at a specified pixel, and the starting pixel becomes a pixel at the current location. The criterion for a pixel in the current position to visit one of the other pixels is the pixel closest to the pixel in the current position. We chose the *K*-D tree to find the nearest pixel in the current position. We constructed a *K*-D tree with the coordinates of the current pixel and unvisited pixels to find and visit the pixel closest to the current pixel. The visited pixel becomes the current location pixel, and the previous location pixel is removed from the *K*-D tree to update the pixel closest to the current location pixel. This process was repeated until all pixels were visited. If the pixel nearest to the current location pixel is itself, it visits the starting pixel and ends route creation.

Figure 12 shows the route obtained by designating a specific pixel as the starting pixel and the object area according to the route. According to the MOED results, the number of routes is equal to the number of detected pixels, and the area of the object may be different for each route. Among the many routes, the Optimal Route that we judge is the route in which the sum of the pixel-to-pixel visit distances is minimum and the object area is maximum, which we define as the optimal route. We connected the pixels according to the Optimal Route and applied Flood Fill. The area of the object obtained according to the Optimal Route is defined as the Optimal Area. To determine the optimal route, we must compare the most optimal route among all possible routes. The condition of the Optimal Route that we determine is that the sum of the visit distances between the pixels should be minimum. Therefore, we determine the route for which the sum of the visit distances between pixels is minimal. A route with the same sum of visit distances between pixels connects pixels and compares the areas to select the route with a larger area. Optimal route determination requires a comparison of all the routes. However, a comparison of all routes presents two problems. First, a *K*-D tree should be constructed with as many pixels as the number of detected pixels when determining a path. Second, the sum of visit distances between pixels is generally the same. In both problems, the amount of computation required is significant. The amount of computation also increases as the number of detected pixels increases; therefore, comparison of all the routes is difficult. Therefore, without comparing all paths, it randomly selects some of the detected pixels and selects the optimal path among them. Randomly selecting a part of the whole is not effective.

We obtained the object’s edge through the MOED and the optimal area through the Optimal Route. The Optimal Area can be better or worse than that of the mask. We need the GT for the object size to determine whether the Optimal Area is improved relative to the mask. However, judging that the Optimal Area is not improved relative to the mask, it can be estimated using cardinality, even if there is no GT for the object size. We compared the cardinality of the Optimal Area and the Mask to validate that the Optimal Area was not improved relative to the mask. The Comparison of Objects Area (COA), an expression for comparing the cardinality of the Optimal Area and the *Mask*, is defined as follows:(3)$$COA=\left\lfloor 100\frac{\left|\mathrm{Optimal}\mathrm{Area}\right|}{\left|Mask\right|}\right\rfloor$$

In COA, *Mask* and Optimal Area represent the number of detected pixels. The ratio of the number of pixels between the *Mask* and Optimal Area is calculated using COA. If the COA is greater than 100, the Optimal Area is equal to or greater than the *Mask* area, implying that the Optimal Area exhibits more pixels than the *Mask*. Therefore, if COA was greater than 100, exciton processing was performed. We judge that the COA is not better than the mask if it is greater than a specific value. Specific COA were identified experimentally.

### 3.3. Exception Processing

Figure 13 shows the flow diagram of Han’s algorithm that detects edges in an image using a Canny Edge Detector and removes pixels outside the mask through a Mask and Bitwise AND operation. The edges detected using the nearest neighbor algorithm were subsequently connected, and a floodfill was applied. The strengths and weaknesses of Han’s algorithm are the Bitwise AND operations. The advantage of the bitwise AND operation is that there is no case in which the object is considerably larger than the mask based on the erasure of the pixels that exist outside the mask. The disadvantage of the bitwise AND operation is that for a large number of detected edges, there is no difference between the results of applying the algorithm and the mask. In other words, Han’s algorithm renders it difficult to identify apprOPRiate or inapprOPRiate cases, regardless of the image applied. Therefore, we used Han’s algorithm for exciton processing. Exception processing handles exceptions for situations in which the proposed algorithm may cause errors. Exception processing is applied in two cases. The first is when the OPR of the edge image acquired through the MOED exceeds the set OPR value. In this case, more pixels were detected outside the mask than inside the mask, and there was a high probability that the Optimal Area would be larger than the mask. The second occurs when the COA exceeds a set value. In this case, the probability of the Optimal Area being larger than that of the mask is high.

## 4. Experiment and Results

We defined six parameters: four used in MOED and two used in VAPOR. The parameters used for the MOED were sigma, eps, minPts, and MCS. The size of the SF was determined using sigma. Points that did not form clusters were removed according to eps and minPts in DBSCAN in the MOED. The MCS was used to determine whether the size of the cluster formed using DBSCAN was apprOPRiate. In addition, we determined that a cluster smaller than the MCS was meaningless noise and removed it. The parameters used for VAPOR were OPR and COA. OPR is the ratio between the pixels inside and outside the mask used to distinguish the normal/abnormal of the detected edge image. The COA is a value for judging whether the Optimal Area is improved relative to the mask. Therefore, if the COA is less than a specific value, the proposed algorithm fails.

This paper proposes an image-processing-based algorithm to improve object segmentation accuracy. The proposed algorithm uses six parameters, and the area of the object changes drastically according to changes in each parameter. Therefore, experimentally determining the value of each parameter to optimize the proposed algorithm is necessary. The duplicate error rate (DER) is the evaluation metric presented in this paper. We measured the DER value between the resulting image and the GT. In addition, to solve the optimization problem using the measured DER, a greedy algorithm was applied that selected the optimal solution based on the current information.

### 4.1. Experimental Environment

The pothole images used for the experiment were considered at an altitude of >40 m from the ground using a SenseFly drone on roads near cities such as Gwangju, Gunsan, Hwasun, and Pyeongtaek in South Korea. We extracted the pothole region using a Mask R-CNN model trained on the pothole data. Because the inference result of the Mask R-CNN was not 100% accurate, verifying the inference result was necessary. However, the identification of an object in an image captured at an altitude of 40 m or more is difficult. In addition, potholes are more difficult to judge because they are smaller than typical objects. There are 65 pothole images remaining after data verification.

Intersection over Union (IoU) is a traditional metric used in computer vision tasks, such as object detection and segmentation, to measure the overlap between two bounding boxes or masks. The IoU is calculated as follows:(4)$$IoU=\frac{\left|GT\cap RESULT\right|}{\left|GT\cup RESULT\right|}$$

We evaluate the performance of the proposed algorithm using verified pothole images and determine the optimal parameter values. In this study, the resulting images were evaluated using the DER evaluation metric. The DER is calculated as follows:(5)$$DER=\frac{\left|{\left(GT\cap RESULT\right)}^{c}\right|}{\left|GT\cup RESULT\right|}=1-IoU$$

DER is the ratio of non-duplicated regions between the resulting images and GT. GT is the actual pothole area, and the RESULT is the resultant image processed according to the proposed algorithm. The DER is a metric to quantify the error between the result and the GT. The DER metric quantifies the discrepancy between the GT and the resultant image, providing an intuitive measure of the performance of an object detection or segmentation algorithm. The resulting images were evaluated by changing the parameters using the DER.

We need to determine the parameters based on the optimal DER by changing the six parameters sigma, eps, minPts, MCS, OPR, and COA to identify the optimal parameters. However, correctly measuring the change in the DER by each variable change owing to Exception Processing is difficult. Therefore, we have discussed the experiments in two parts (Section 4.2 and Section 4.3): with and without exception processing. In other words, exception processing was not applied when obtaining sigma, eps, minPts, and MCS (Section 4.2).

### 4.2. Experimental Optimization of MOED Parameters

MOED is an algorithm for extracting only the edges of actual potholes in an image and consists of two-step edge detection and noise filtering. First, the edge detection step detects edges using an SF and an LF. Second, the noise-filtering step removes noise using the DBSCAN and MCS equations. To ensure the optimal performance of the MOED algorithm, experimentally determining the optimum values of its parameters is necessary.

#### 4.2.1. Experimental Optimization of SF

The SF in the MOED algorithm uses the sigma parameter to set the standard deviation of the Gaussian filter. As the sigma value increases, the degree of blurring of the filter also increases. Therefore, we experimentally determine the optimal sigma value.

Figure 14 shows the experimental results for determining the optimal sigma parameters. We used a boxplot to evaluate how the DER changes according to sigma variation. From the general distribution of the data in Figure 14, the DER increases based on sigma 4. It indicates that the data points are closely clustered around that sigma value 4, suggesting that the data is relatively consistent and less spread out, which can be desirable in DER-determined. Therefore, we set the optimal sigma value to 4.

#### 4.2.2. Experimental Optimization of DBSCAN

DBSCAN uses the eps and minPts parameters to filter noise. Algorithms use these parameter values to determine the proximity of data points to find clusters. Therefore, determining apprOPRiate parameter values to obtain the best results is important. The optimal values of eps and minPts were experimentally determined.

Figure 15 and Figure 16 show the experimental results of determining the optimal eps and minPt parameters. Figure 15 shows that when the eps was 2, the DER was the lowest and densest. It indicates that the clusters formed were well-separated and had minimum dissimilarity or error. The clusters formed are well-separated and have minimum dissimilarity or error, implying that the clustering is well-performed. Figure 16 shows that when the minPt values were 11 and 12, the DER values were generally lower than the other values. When minPts was 11, unlike when minPts was 12, an outlier was observed. However, based on the data distribution, the DER value was less than 12. It shows that by setting the minPts parameter to either 11, the DER generally decreased compared to other values. It indicates that these minPts values resulted in more accurate clustering. Therefore, we set the value of eps to two and the value of minPts to 11.

#### 4.2.3. Experimental Optimization of MCS Equation

MCS was used to verify the significance of the clusters formed using DBSCAN. Our analysis was significant only when the cluster formed using DBSCAN had more than a specific ratio of the total number of pixels in the image. Therefore, we experimentally determined the *coeff*icients that optimize the equation.

Figure 17 shows the experimental results of determining the optimal sigma parameters. The data distribution was analyzed in the previous experiments (Section 4.2.1 and Section 4.2.2) to determine the optimal parameters. However, as shown in Figure 17, the MCS did not change significantly in DER under the fluctuation of MCS. Therefore, the application of this method to determine the optimal parameters by observing the data distribution is difficult. We determined the optimal MCS by examining the effect of the elapsed time of the algorithm on the DER, along with the MCS.

A regression analysis is a statistical procedure used to calculate the effect of an independent variable on a dependent variable [33]. We now describe certain important indicators used in the regression. *R*^2^ is the *coeff*icient of determination that indicates how effectively independent variables predict the dependent variable. The range is between 0 and 1; the closer the value is to 1, the better the independent variable predicts the dependent variable [34]. The *p*-value is used to determine the statistical significance of each predictor, and a statistically significant case is below a certain threshold (usually ~0.05) [35]. The terms significance level and significance F are used interchangeably. Depending on the researcher, the criteria for accepting or rejecting the null hypothesis differ; however, if the significance level is less than 0.05, it is rejected [36]. Therefore, as shown in Table 1, we used multiple regression analysis to evaluate the impact of the two variables on a single variable.

Table 1 shows the results of the multiple regression analysis. According to the results in Table 1, the significance F-value is less than 0.05, the adjusted *R*^2^ is approximately 0.65, and the *p*-values of MCS and elapsed time are less than 0.05. This indicates a statistically significant correlation between MCS, the elapsed time, and DER.

In conclusion, the optimal MCS value was determined to be 21 based on an analysis of these two factors. First, as the MCS increased, the DER decreased. However, when the MCS value was set to 21, the variation in the DER was minimal. Second, as the MCS increased, the elapsed time also increased. Therefore, considering the tradeoff relationship between MCS and the elapsed time, we set the optimal MCS value to 21.

### 4.3. Experimental Optimization of VAPOR Parameters

VAPOR evaluates the MOED results using the COR and COA equations. Because the two equations yield values ranging from 0 to 100, the optimal values must be determined experimentally to improve verification performance.

Figure 18 presents the DER results based on the quantization level. OPR and COA are parameters with values between 1 and 100. Because the number of possible combinations was 10,000, determining an optimal value was difficult. Therefore, we determined the values suitable for the proposed algorithm through quantization. The quantization level was gradually increased to identify values distributed in the lowest range. The quantization levels ranged from 3 to 14. The data distribution was divided into integer units at a quantization level of 14. Further division of the quantization level by 15 or more complicates the data distribution. Therefore, the number of quantization-level subdivisions was limited to 14 for simplicity.

We determined the values optimized for the proposed algorithm to be in the range of 44 to 45 at a quantization level of 14. The DER of OPR 19, COA 36, one of the values in the DER 44–45 range, was 44.43%. The average DER of the instance segmentation of the Mask R-CNN was 57.23%, and the DER of our proposed algorithm was 44.4329%. Therefore, our proposed algorithm is improved by approximately 13% relative to Mask.

Figure 19 shows an example of a segmentation comparison between the proposed algorithm and the mask. As shown in the last row of Figure 19, the proposed algorithm depends on the results of the Mask R-CNN. Therefore, if the inference result of the Mask R-CNN is incorrect, the proposed algorithm cannot obtain an apprOPRiate outcome.

Figure 20 shows the results of pothole contours on images using ACMs. ACMs show relatively inaccurate relative to true contours in the case of portholes. We estimate that roughness and cracks in the asphalt road cause these discrepancies.

## 5. Conclusions

In this study, we proposed an algorithm to improve the accuracy of tiny object segmentation for accurate asphalt pavement pothole detection. The proposed algorithm comprises a three-step procedure: MOED, VAPOR, and Exception Processing. This approach effectively extracts pothole edges, validates the results, and apprOPRiately handles any abnormalities detected. Unlike object detection, instance segmentation shows objects in an image in more detail in pixel units than in bounding boxes. In aerial images, the actual area occupied by the pixels in the image increases in proportion to the altitude captured by the UAV; thus, the segmentation accuracy of the instance segmentation model is more important than that of the object detection model. When shooting at a relatively high altitude using a UAV, the size of the pixels occupied by each object decreases; thus, the accuracy of instance segmentation is low. Methods to improve the performance of these instance segmentation models include increasing the inference performance through additional training or postprocessing. Among the two strategies, we selected a method to improve inference performance through post-processing. Therefore, in this study, a post-processing algorithm to increase segmentation accuracy was proposed and evaluated experimentally. The error rate of our proposed algorithm was reduced by approximately 13% compared with when the proposed algorithm was not applied. However, the parameter combination used in our proposed algorithm may not be optimal. The proposed algorithm requires six parameters, each of which ranges from 1 to 100, and the number of possible combinations is H6100 = 1,609,344,100. Because one combination consists of 65 images and processing each combination using the proposed algorithm takes approximately 17 s on average, a simple calculation takes approximately 75,996 h and 3166 days. Such an experiment, which requires a long time, is practically difficult to implement. Therefore, we applied the greedy algorithm and confirmed the results in approximately 1100 h. However, if the inference result of the Mask R-CNN is incorrect, the proposed algorithm cannot reach an apprOPRiate outcome, which is a limitation.

In the future, we will apply the proposed algorithm to objects other than potholes to evaluate its performance and determine the apprOPRiate parameters for each object. In addition, when a more accurate object area is required, such as for a pothole, the area of the object can be estimated more effectively using the proposed algorithm.

## Figures and Tables

**Figure 1 sensors-23-05851-f001:**
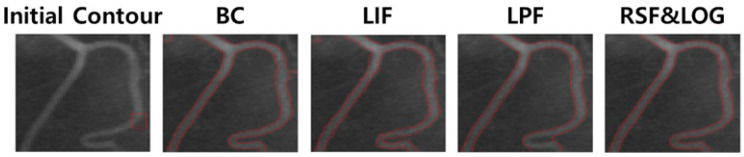
The results of each ACMs to segment medical image.

**Figure 2 sensors-23-05851-f002:**
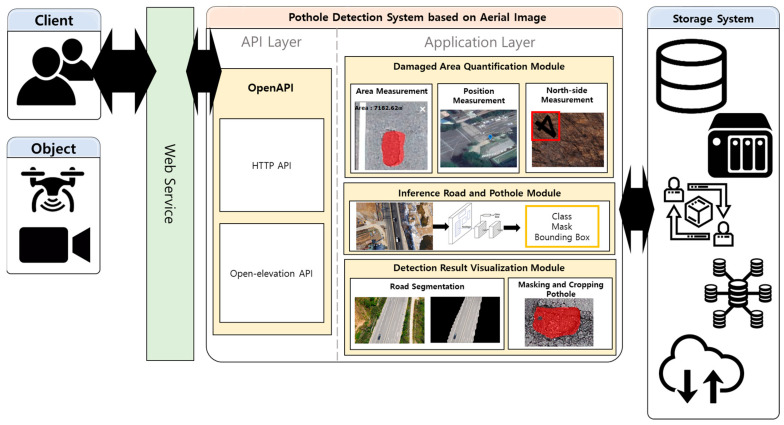
Legacy pothole detection system.

**Figure 3 sensors-23-05851-f003:**
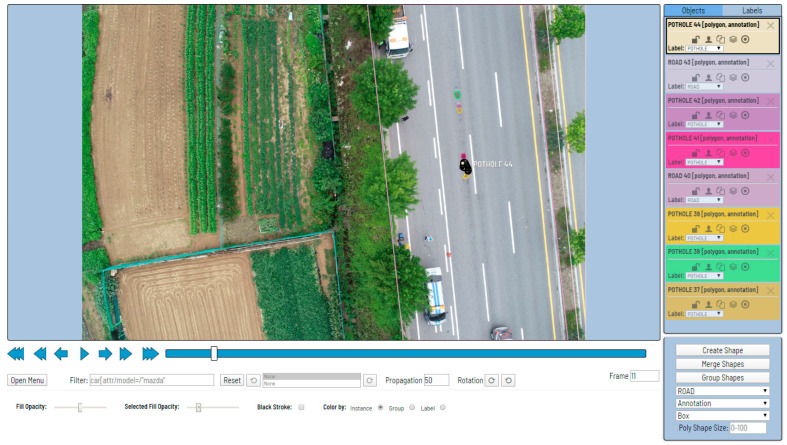
Example of annotation and labeling using CVAT.

**Figure 4 sensors-23-05851-f004:**
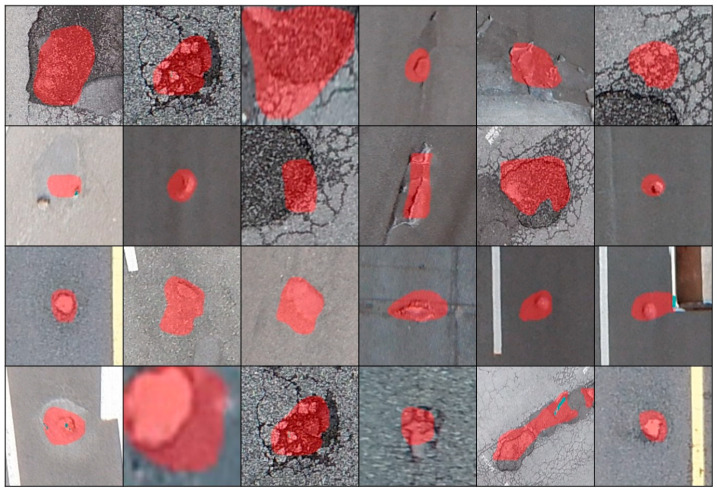
Pothole inference result using Mask R-CNN.

**Figure 5 sensors-23-05851-f005:**
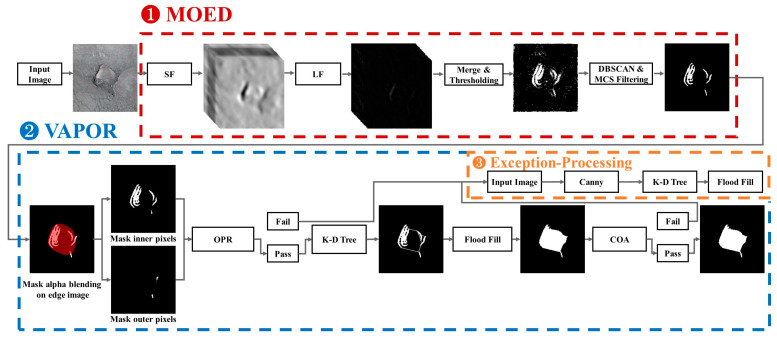
Flow diagram of proposed algorithm: (1) Extracting the edges of actual potholes in the image. (2) Normality or abnormality of MOED results, applying a *K*-D tree for normal results. (3) Application of Han’s image processing-based algorithm into the exception processing method.

**Figure 6 sensors-23-05851-f006:**
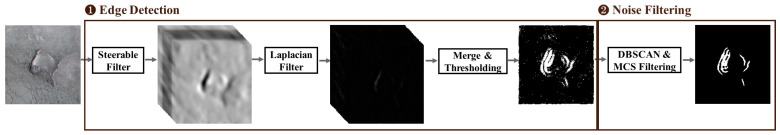
Visualization of an example flow of MOED.

**Figure 7 sensors-23-05851-f007:**
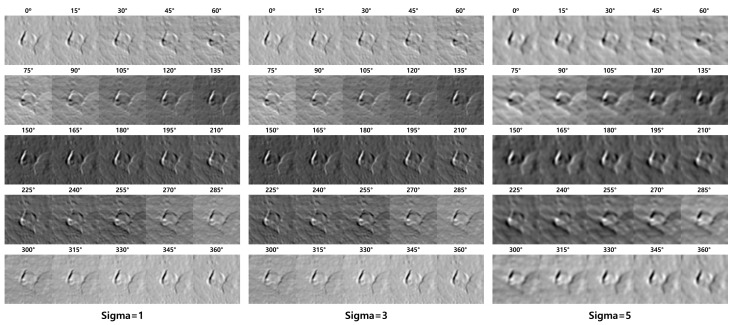
Result images according to parameter sigma and theta values in SF.

**Figure 8 sensors-23-05851-f008:**
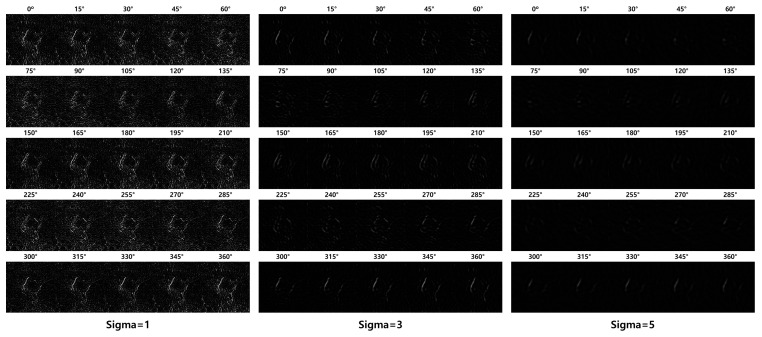
Result of applying LF to the SF result.

**Figure 9 sensors-23-05851-f009:**
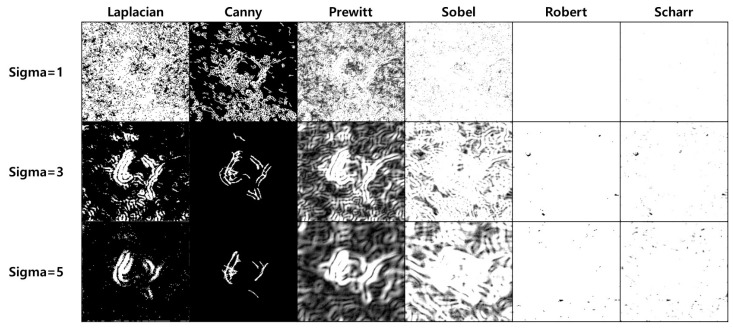
Example of merging after applying an edge detection filter to the SF result.

**Figure 10 sensors-23-05851-f010:**
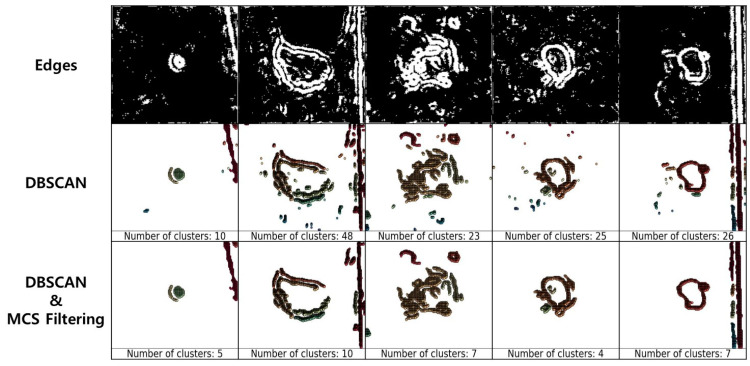
Image denoising using DBSCAN and MCS filter.

**Figure 11 sensors-23-05851-f011:**
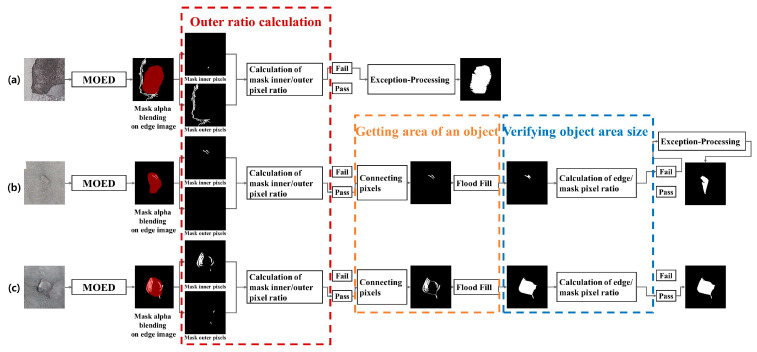
Visualization of an example flow of VAPOR: (**a**) Case of failure owing to mismatch with edge image’s inner/outer pixel ratio. (**b**) Case of failure owing to mismatch with the minimum size compared with the mask image. (**c**) Case of success of the proposed algorithm.

**Figure 12 sensors-23-05851-f012:**
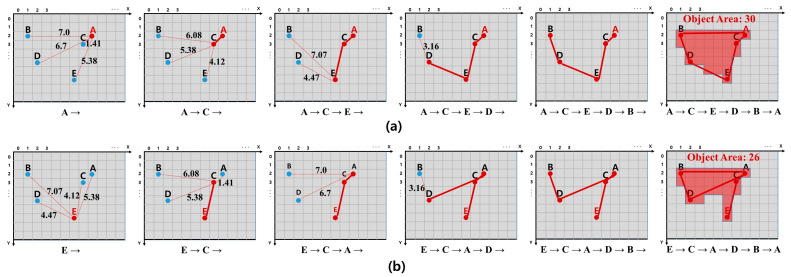
Object area difference when the starting points are different: (**a**) Start and end at Pixel A. (**b**) Start and end at Pixel E.

**Figure 13 sensors-23-05851-f013:**
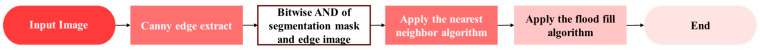
Flow diagram of Han’s improvement of object segmentation accuracy algorithm.

**Figure 14 sensors-23-05851-f014:**
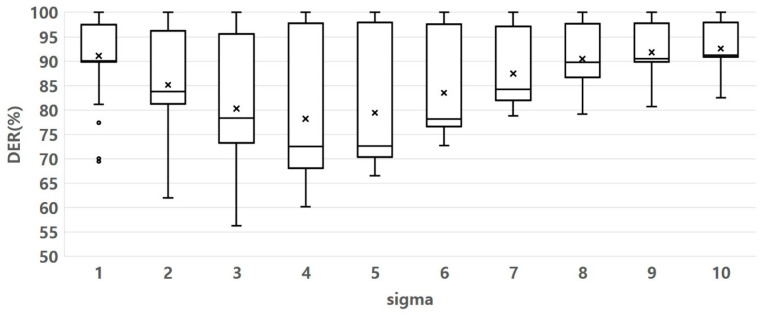
Sigma parameter variation effect on DER value. Other parameter fluctuation range: eps 1–16, minPts 10–100, MCS 10–100.

**Figure 15 sensors-23-05851-f015:**
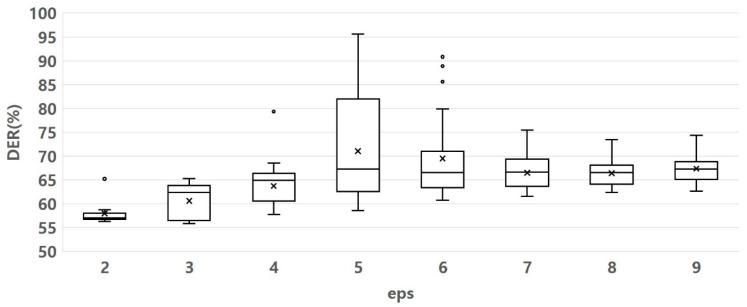
eps parameter variation effect on DER value. Other parameter fluctuation range: sigma 4, minPts 10–100, MCS 10–100.

**Figure 16 sensors-23-05851-f016:**
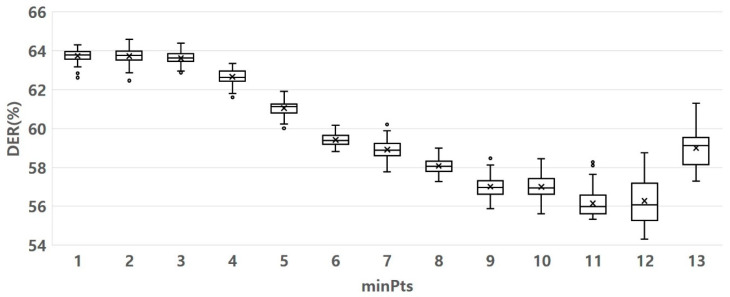
minPts parameter variation effect on DER value. Other parameter fluctuation range: sigma 4, eps 2, MCS 30—84.

**Figure 17 sensors-23-05851-f017:**
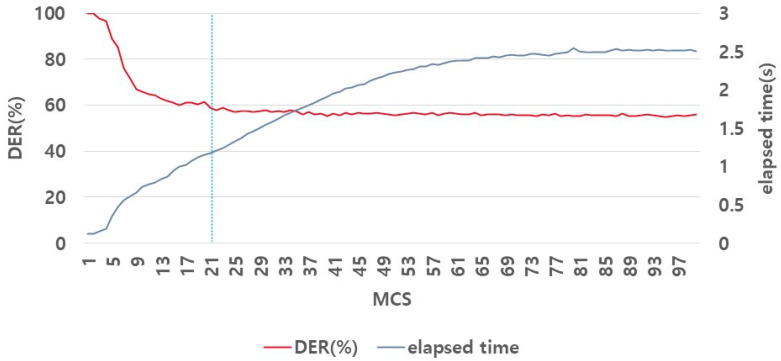
MCS parameter variation effect on DER value. Other parameter fluctuation range: Sigma 4, eps 2, minPts 11.

**Figure 18 sensors-23-05851-f018:**
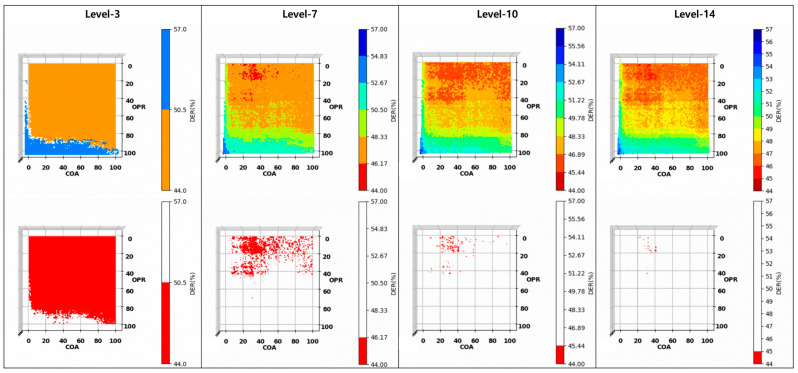
Analysis of the impact of OPR and COA on DER performance through quantization techniques.

**Figure 19 sensors-23-05851-f019:**
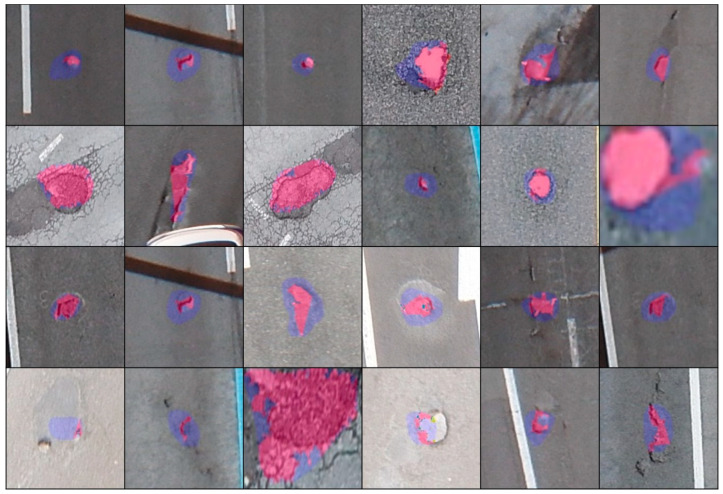
Example of segmentation comparison between the proposed algorithm (shown in pink) and Mask R-CNN instance segmentation (shown in blue) on the high-altitude captured images.

**Figure 20 sensors-23-05851-f020:**
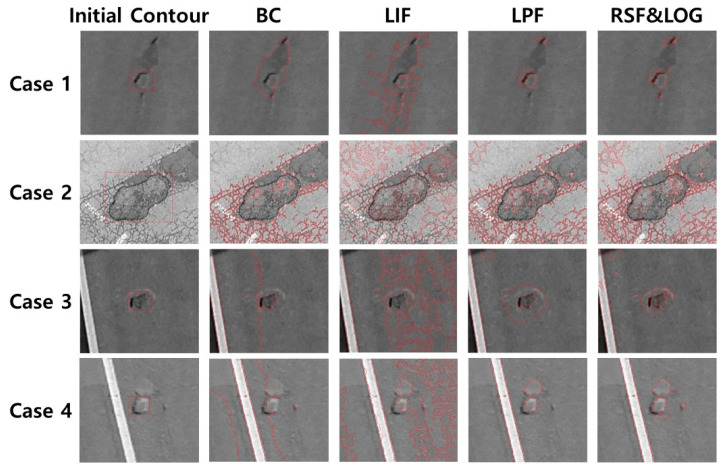
Pothole segment result using ACMs.

**Table 1 sensors-23-05851-t001:** Effect analysis of MCS and elapsed time on DER using multiple regression analysis.

Regression Statistics								
Multiple R		0.836881						
R Square		0.70037						
Adjusted R Square		0.694192						
Standard Error		5.318337						
Observations		100						
ANOVA		Degree of freedom		Sum of Squares	Mean Square	F	Significance F	
Regression		2		6413.057	3206.529	113.3661	4.12 × 10^−26^	
Residual		97		2743.617	28.28471			
Total		99		9156.674				
	*Coeff*icients	Standard Error	t Stat	*p*-value	Lower 95%	Upper 95%	Lower 95.0%	Upper 95.0%
Intercept	84.68844	1.780375	47.568	4.91 × 10^−69^	81.155	88.222	81.155	88.222
MCS	0.277518	0.048762	5.6912	1.34× 10^−7^	0.1807	0.3743	0.18079	0.3743
Elapsed time	−20.7874	1.972863	−10.54	9.34× 10^−18^	−24.7	−16.872	−24.703	−16.87

## Data Availability

Data sharing not applicable.

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
