# Peer review of "Improvement of Tiny Object Segmentation Accuracy in Aerial Images for Asphalt Pavement Pothole Detection"

_sensors, 2023, doi:10.3390/s23135851_

Round 1

Reviewer 1 Report

The manuscript is written in an exhaustive manner and is verbose. Try to reduce the verbosity as it distracts the reader. The introduction section is quite long. Also, it would be great if other similar works in literature will be mentioned except only Han's work. So that the recent state-of-art can be established. 

Author Response

We are indebted to the reviewers for their valuable comments which were extremely helpful in revising our manuscript. However, after considering their comments, we thought that our manuscript must be re-wrote in all section. Finally, we reconstructed and wrote all parts of our manuscript. We have given due considerations to each and every comment in preparing the revised manuscript.

Reviewer 2 Report

This paper focuses on improvement of tiny object segmentation accuracy in aerial images for asphalt pavement pothole detection. In general, the topic is interesting and some comments are given as follow:

1. The work done in the article should be described more specifically in the first part of the article.

2. The innovation points of this article should be clearly listed.

3. The difference between the proposed evaluation indicator DER and traditional evaluation indicators should be introduced.

4. In the experiment, traditional indicators should be compared with the proposed DER, or the reason why DER replaces traditional evaluation indicators should be introduced.

5. The parameter values of the selected 6 parameters should be explained in more detail.

6. Image 10 is not clear enough, and the text in the image is blurry.

7. Other segmentation methods are suggested to add in the literature review, such as active contour model, An overview of intelligent image segmentation using active contour models, Intelligence & Robotics 3 (1), 23-55.

Minor editing of English language is required.

Author Response

(The authors gave the same response as above.)

Reviewer 3 Report

This paper makes an in-depth study on instance segmentation. The authors propose an image processing-based algorithm for detecting potholes on road surfaces using unmanned aerial vehicles (UAVs), and carrying out experimental evaluation. The flow diagram of the proposed algorithm consists of three steps: Multiple Orientation Edge Detector (MOED), VAlidation Phase Of Results (VAPOR), and Exception Processing.  Finally, the limitations of the study are discussed.

There are some problems that need to be improved in this paper:

1、In the part of Introduction, should introduce your work and innovation step by step, so you need to supplement the progression logic. For example, why was the algorithm proposed by Han selected for improvement, and what are the key defects of his algorithm that need to be improved?

2、Lack of summary. It is best to summarize the algorithm innovation points and calculation results in the first part.

3、In the part of Related Work, the introduction should also be relevant to the domain, rather than introducing DBSCAN and so on directly from the beginning, which can be remarkably abrupt (part 2.1-2.3).

4、The proposed algorithm should be more explicit. The composition of the algorithm should be mentioned at the beginning and end of the article, rather than limited to the third part.

Generally, I think the quality of the paper is good but need to improve.

Author Response

(The authors gave the same response as above.)

Reviewer 4 Report

The proposed algorithm addresses the limitations of previous methods and offers several advantages, including wider coverage, enhanced accessibility to difficult locations, and increased operator safety. They experimentally evaluated the performance of real data in various regions of South Korea. Overall, the submission lacks innovations and is not written well. Followings are some suggestions for improvements.

1. In the abstract, the contributions are not apparent in quantity when compared to some of the SOTAs.

2. In related works, the potholes detection methods, except Han's [9], are not mentioned at all. So, this submission is too limited to see other related works.

3. Their modifications to [9] are not so significant. Basically, this submission still adopted the same main structure and did some preprocessing, like filters and edge detection. So, the innovation parts are quite weak.

4. Comparisons with some SOTAs should be made in the experiments.

5. Title format of references needs to be uniform. 

Typos and grammar errors should be corrected.

Author Response

(The authors gave the same response as above.)

Reviewer 5 Report

This is an interesting paper, generally well-written and structured with alot of detail etc. It does seem rather lengthy and whilst the background and rationale are very well covered it seems to take a long time to get to the results, which are a little bit of a come down given the build up to this point. Initially the paper seems to suggest that there is an approach that is going to lead on to the demonstration of a practical result - which at the end - is not quite the case as there appear to be some difficulties in achieving this. The potential of the approach and practical application are clear and comparisons with other ways to do this are well explained at the outset. Clearly, it is not an easy problem to solve, although development and refinement of an existing approach seems to have potential.  To this end, and given that the paper is quite lengthy and detailed, I wonder if there is some merit in redrafting and structuring the paper in such a way as to help convey the theory, the experimentation and preliminary results that then highlight some of the next steps in addressing the difficulties? I also felt that some of the illustrations of the approach could be annotated more to help the reader understand the outcomes and what each illustration is showing the reader. This could even include visuals as to what the method is doing and once applied the success/failure etc. To summarise, for the reader who is perhaps trying to understand how this complex approach works in practice, and how well it performs practically, maybe a little rethink of how the 'technical' aspects are communicated visually could prove more convincing for the reader of how well this approach performs. The technical aspects are all very well documented and detailed, but illustrating and annotating the outcomes of each step in the approach to help extract the information from the drone imagery that is needed for this application, all seems a little theoretical, and the limitations/solutions achievable, would perhaps help the reader to see more the potential and appreciate the difficulties. I think this would put a more positive slant on the paper also. The conclusions do not - for me - quite reflect the Abstract.

Very good.  Minor edits only needed as a result of a final read through.

Author Response

(The authors gave the same response as above.)

Round 2

Reviewer 2 Report

The authors have addressed all my comments.

Minor editing of English language is required.

Author Response

We are indebted to the reviewer for their valuable comments which were extremely helpful in revising our manuscript. We really appreciate your help.
Thank you.

Reviewer 4 Report

Still, there exist format problems in Table 1 and Fig. 20.

The submission maybe accepted after corrections.

Need to check the fluency in English.

Author Response

(The authors gave the same response as above.)

Reviewer 5 Report

The paper has very clearly undergone substantial editing of the English and the structure of the paper, as well as the illustrations. This is greatly improved and makes the paper much more accessible to the reader.

Author Response

(The authors gave the same response as above.)
